# Bermudagrass Responses and Tolerance to Salt Stress by the Physiological, Molecular Mechanisms and Proteomic Perspectives of Salinity Adaptation

Maryam Noor, Ji-Biao Fan, Jing-Xue Zhang, Chuan-Jie Zhang, Sheng-Nan Sun, Lu Gan and Xue-Bing Yan *

College of Animal Science and Technology, Yangzhou University, Yangzhou 225009, China
* Correspondence: yxbbjzz@163.com

**Abstract:** *Cynodon dactylon* (L.) Pers. (commonly known as bermudagrass) is a member of the family *Poaceae*. It is a C4 grass that can grow annually and perennially with clone reproduction and seed-setting. It is not only used as forage but also as a weed in many crops. It grows along roadsides, in barren lands, irrigated lands, and seacoasts, where soil salinity is a major problem. Although bermudagrass is distributed worldwide, it shows limited growth under saline conditions. Under salt stress, the whole-plant growth is disturbed and the forage quality is compromised morphologically. At the physiological level, shoot development is affected owing to the resultant oxidative stress, although the total rate of photosynthesis is not greatly affected. Biochemical changes include a change in the $K^+/Na^+$ ratio; antioxidant enzymes such as superoxide dismutase and peroxide dismutase increase, while catalase activity slows down. The anatomical changes are visible as salt crystals on the leaf surface and salt glands on the mesophyll surface. In this paper, the morphological, physiological, biochemical, and proteomic mechanisms of bermudagrass under salt stress are discussed, drawing a study of several genes such as *ECA4*, *RAN1*, *MHX1*, *psbA1*, *psbB1*, *psbP*, and *psbY* at the molecular level. Therefore, the current review aims to understand how bermudagrass grows and adapts well under salt conditions.

**Keywords:** *Cynodon dactylon*; growth; abiotic stress; salt tolerance; adaptation; proteomic study



## 1. Introduction

Elevated soil salinity levels interfere with global agricultural production by causing osmotic stress, ionic imbalance, and specific ion toxicity, and hence affect worldwide agricultural production. Salinity impacts almost 20% of global cultivated lands and half of the total irrigated lands [1]. Nearly 800 million hectares (mha) are reported to be salt-affected globally [2]. In addition, saline soil affects more than 45 mha of irrigated land; among this, ~1.5 mha of land requires reclamation [3]. There are two main types of salinization, i.e., primary salinization, caused naturally by excessive evaporation, high temperature, and decreased precipitation facilitated by climate change, whereas secondary salinization, in most areas, arises due to human activities such as poor irrigation methods and the use of snow-melting salty water in winter in some areas [4]. Saline conditions have also been reported to limit turf grasses' growth and development, leading to their poor growth and quality [5]. Salinity resulted in a 50% reduction in the dry weight of shoots in 35 bermudagrass cultivars [6]. However, using salt-tolerant grass species can diminish the adverse effects of salinity by following a series of morphological, physiological, and biochemical mechanisms (Figure 1) [7]. Halophytes have been known for hundreds of years as highly salt-tolerant plants and are considered as best germplasm for saline soil. Salt-tolerant genes were isolated from different halophytes (*Aeluropus*, *Atriplex*, *Suaeda*, and *Cakile*) and incorporated into crop plants for developing salt-tolerant species. Halophytes follow different mechanisms against salt stress, such as the excretion of $Na^+$ ions, ion compartmentalization, the secretion of salt from glands, reductions in $Na^+$ influx, and changes in membrane composition [3]. Bermudagrass

(*Cynodon dactylon*) belongs to the family Poaceae, a type of perennial herbaceous grass with well-developed rhizomes and stolons [8]. It is extensively used as a warm-season grass, for forage and ecological restoration in tropics, subtropics, and warm–temperate and arid regions. It was demonstrated that *Cynodon* sp. is a highly saline-tolerant turf grass; however, the tolerance patterns vary within the same species [9]. In saline soil and domestic water, salt-tolerant bermudagrass species can produce appropriate turf quality [7] and have outstanding surviving potential under saline–alkali domains [10]. Bermudagrass cultivars exhibit significant genetic variation in terms of salinity tolerance [11,12]. Under high salinity, sodium ($Na^+$) content is higher in the leaves and roots of bermudagrass, which competes with the potassium ($K^+$) content in these parts and decreases its uptake (Figure 2) [13]. A limited study was conducted to explain the mechanisms of the regulation of $K^+$ and $Na^+$ balances in bermudagrass. They stated that the excretion of $Na^+$ was negatively associated with its accumulation in the leaves, but positively correlated with salinity tolerance [6]. In bermudagrass *Cynodon* sp., the activity of the salt exclusion process was weaker; however, it was highly salt tolerant compared to other grasses. At the same time, it was recognized that the transport of selective ions such as $Na^+$ and $K^+$ in bermudagrass was more active than in other grasses. It was suggested that the transportation of selective ions may be a mechanism for the regulation of ions under saline conditions (Figure 3). However, this study was based on only one bermudagrass cultivar and lacked comparative research between varieties of salinity-tolerant genotypes [14]. However, the appropriate understanding of the mechanism by which *Cynodon* sp. can lessen $Na^+$ accumulation and maintain the constancy of $K^+$ in leaves at the whole-plant level under high salinity levels remains unclear. The most salt-tolerant genotypes of bermudagrass were "Tifdwarf" and "Tifgreen", while the most salt-sensitive genotypes recorded were "common" and "Ormond" [15,16].

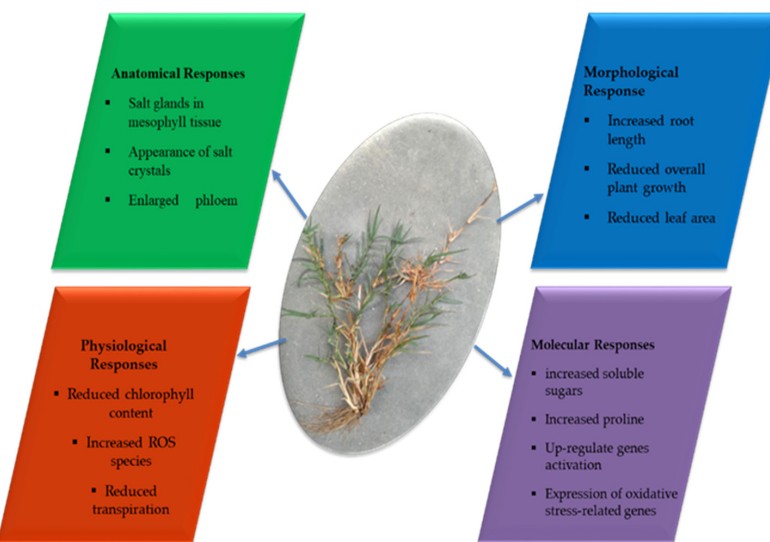

**Figure 1.** Impacts of salt stress on bermudagrass [4–7].

## 2. Effects of Salt Stress on Bermudagrass

Considerable variation is found between bermudagrass cultivars in their adaptation and tolerance towards saline environments [11]. Salinity stress causes the leaf color to be a little darker than normal or bluish-green, and the narrow blades become stiffer with more pointed edges. Leaf size is reduced and wide edges are visible, indicating cell-size reduction (Figure 1). Reduced intracellular spaces, strengthened palisade parenchyma, and thickened cuticles, either alone or in combination, may result in a darker green color of the leaves [16,17]. Moreover, wilting can occur if a large quantity of saline water is supplied, which results in asymmetrical shoot growth. If toxicity from particular ions occurs, then necrotic spots might appear on the surface of the leaves. The shoot color gradually appears

darker green and the leaves become wilted as the salt level increases (see Figure 1) [18]. In addition, high salinity leads to leaf tips burning, which finally spreads downward and covers the entire leaf surface. At this stage, the growth of the shoot declines and stunted growth occurs. As the salt level rises, the growth of the root's declines, though in some cases, the roots may still enlarge. If protective measures are not undertaken timely, the growth of the grass may be compromised, the shoot density may decrease, and consequently, the turfgrass may appear thinner, leaving asymmetrical-looking turfgrass patches [19].

### 3. Salt-Stress Responses in Bermudagrass

In bermudagrass, salt stress results in (1) osmotic stress, (2) ionic stress, and (3) oxidative stress [20]. Osmotic stress is a rapid mechanism that limits water uptake and is responsible for the closure of stomata, reducing cell expansion and division. On the other hand, ionic stress creates an ionic imbalance, disrupts homeostasis and the cellular functions of ions, and, ultimately, causes leaf senescence. Ionic disequilibrium results in cell membrane leakage, alterations in mineral distribution, reduced turgor pressure, and even plant death. In addition, oxidative stress results in the over-production of reactive oxygen species (ROS) such as singlet oxygen ($^1O_2$), hydroxyl radical (OH), and hydrogen peroxide ($H_2O_2$). Normally, ROS are generated in plants, but generally, there exists a balance between their production and scavenging by different antioxidant enzymes. ROS have an important function in cell signal transduction, but their excessive accumulation causes oxidative damage to the membrane proteins, lipids, and nucleic acid [9]. The regulation mechanism of $Na^+$ and $K^+$ balance in *C. dactylon* is under debate. It can tolerate salt stress by excreting $Na^+$ out of the cells, halting its uptake, and ensuring its translocation into vacuoles within cells with the aid of mechanisms (ion pumping, active transport, ion exchange) [21]. Members of the family *Poaceae*, including bermudagrass, are salt-secreting grasses with the characteristic of bicellular salt glands. As $Na^+$ uptake is reduced, this leads to its decreased accumulation within the vacuole as well as its associated injuries [22]. Despite having excellent salt tolerance, bermudagrass exhibits a poor capacity for salt excretion as compared to *Z. japonica*; however, it was also observed that in bermudagrass, the selective transport capability of $Na^+$ and $K^+$ was more powerful than in *Z. japonica*. Still, this research lacked a comparative examination between various salt-tolerant cultivars [14]. However, the mechanisms behind the decreased $Na^+$ uptake and increased $K^+$ retention in bermudagrass leaves under saline stress are still unclear [23]. In salty conditions, an increase in $Na^+$ and a decline in the content of $K^+$ in the growth mechanism of *C. dactylon* were observed (Figure 3) [24]. The $Ca^+$ content in shoot sections varied under diverse salt levels in the "Satiri" and "Tifdwarf" cultivars of *C. dactylon*.

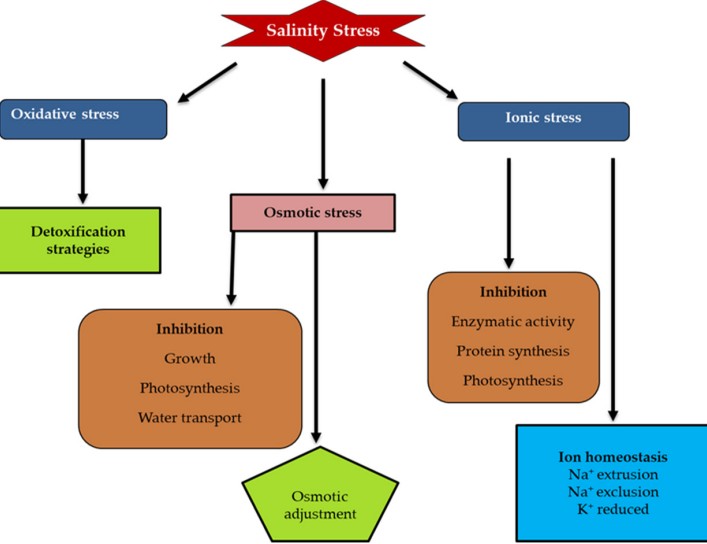

**Figure 2.** Types of salinity stress responses in bermudagrass [22–25].

Calcium is an essential element of a plant's cell wall and cell membrane. It contributes to the structural and functional integrity of the membrane and cell wall enzyme activities and regulates ion transport and selectivity. Under stressed conditions, the main function of $Ca^+$ is acting as a secondary messenger. The external stimuli or stress signal opens the $Ca^+$ channel, releases it from the vacuole, binds with proteins (i.e., calmodulin), and regulates gene expression and cell metabolism. However, there are several research studies on the significant inhibition of the uptake of $Ca^+$ under salt stress [25]. In these studies, the *C. dactylon* genotypes "Satiri" and "Tifdwarf" exhibited enhanced $Ca^+$ accumulation in shoots. In osmotic regulation, the regulators could be organic or inorganic substances, which increased to different levels under salt stress. Organic regulators include free amino acids, mainly proline; betaine, mainly glycine betaine; and soluble carbohydrates. Proline has a hydrophobic end that binds to proteins, while the hydrophilic end binds to water molecules. The proteins attached to proline can attach with more water molecules to prevent denaturation. So, proline acts as an antioxidant to scavenge ROS and regulates cell osmotic pressure by preventing water loss under salt stress. Glycine betaine also acts as an osmotic regulator by maintaining membrane integrity and enzyme activity under salt stress [9]. Recently, a large number of nutrient substances, signaling molecules, and metabolites were shown to develop the tolerance of bermudagrass against salt (Figure 3) [26]. In addition to enhanced $K^+$ concentration in shoots, $Na^+$ elimination from the leaf surface was also recorded under salinity [27]. The findings suggested that, unlike other grasses, bermudagrass used $Na^+$ removal to cope with salinity (Figure 3) [28]. The *C. dactylon* maintained fairly stable levels of $K^+$ with increased salinity conditions. Normally, ratios of $K^+/Na^+$ in the shoot portion of most grasses decrease due to increased salt stress, but in *C. dactylon*, that ratio remained higher at a high salinity level, demonstrating the selectivity of shoots for $K^+$ over $Na^+$ [29].

In bermudagrass, osmotic adjustment can be facilitated by the redistribution and substitution of ions (Figure 2) [30,31]. Concentrations of total inorganic ions did not change more with increasing amounts of $CaCl_2$ and NaCl. Although, some mechanisms of substitution may be present to avoid the production of noxious mineral elements. The alteration in ion concentrations in the cortical regions of roots may be due to ions in the free spaces of a cell. Changes in the concentration of ions might be due to different free spaces. The decrease in concentrations of magnesium ($Mg^{2+}$) and calcium ($Ca^{2+}$) in plants developed with extra potassium sulfate ($K_2SO_4$) is not surprising because when the supply of $K^+$ is large, then the uptake and transport of other cations are diminished, as in Figure 3 [32,33]. The strategy of bermudagrass to cope with salinity is an increase in root growth with a lessened top growth, which may enable it to survive osmotic and nutritional stresses caused by salinity (Figure 2). Under salt stress, bermudagrass can be categorized as an ion regulator by maintaining low levels of $Na^+$ and $Cl^-$ in vessels in roots, while maintaining high $K^+/Na^+$ ratios in leaves and having extremely active and selective salt glands [29].

## 4. Morphological Responses

Bermudagrass is considered an outstanding grass to re-vegetate since it adapts a sequence of morphological responses under saline and saline–sodic soil conditions (Figure 1) [10]. The morphological indicators used in evaluating salt stress are generally the plant biomass attributes such as root, shoot, and leaf weights as well as their lengths and diameters, as they represent the overall plant growth under salinity. Normally, biomass decreases as the salt level increases but the degree of reduction depends on the plant species [22]. Salt stress causes a reduction in overall turf quality, in the number of stolons, and in both the weight and height of shoots, whereas an increase in the length and numbers of the roots and root/shoot ratio of salt-tolerant bermudagrass genotypes was also observed (Figure 1) [13,16]. The lengths of the roots and shoots of *Cynodon* sp. were significantly improved in NaCl solution, but reduced when NaCl concentration was further increased [34]. The root mass of bermudagrass cultivars Tifway, Tifdwarf, and Tifgreen were positively associated with

salinity tolerance [35]. Using four different *Cynodon* genotypes, namely Dacca, Khabbal, Tifway, and Tifdwarf, the research revealed that the number of roots and the dry weight significantly decreased in all these genotypes at five diverse salt-stress levels [13]. It was found that salt stress may certainly reduce the root's fresh and dry weights. It was revealed that different results were created by diverse levels of salt tolerance in different bermudagrass genotypes and the treatments of salt used in that experiment [36]. A common salt-stress-tolerant population was used, and it was found that the root, stem, and leaves changed in morphology. The research conclusions showed that the morphological changes influenced by salt stress were discriminatory according to the plant's tolerance against salinity and its ability to save water [37,38]. It also reduces the $K^+$ concentration but enhances the concentrations of $Na^+$ and $Cl^-$ in shoots and the stolon area of grass [15,39]. In salt conditions, leaf area is also a fundamental growth attribute contributing to whole-plant growth. However, in salt-adaptive cultivars, the leaf area is not greatly affected under saline conditions [40]. Salinity stress substantially affects root length, but this possibly differs with plant or genotype. In general, *Cynodon dactylon* with a high root number and length under salinity is recorded as one of the most salt-tolerant cultivars (Figure 1) [41,42].

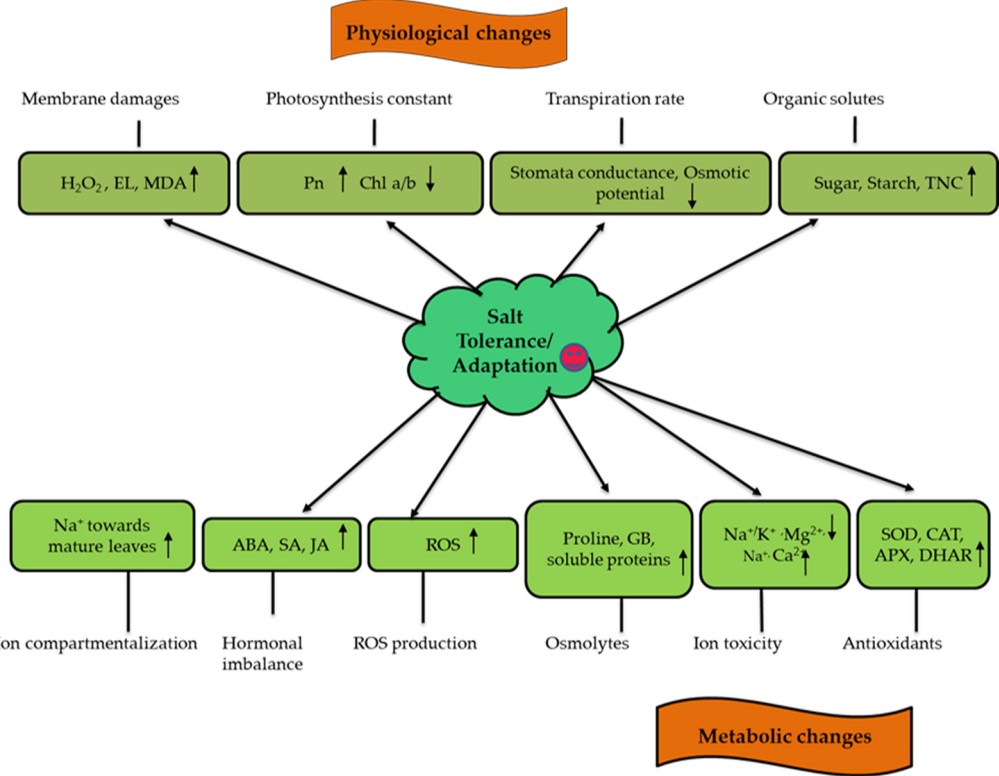

**Figure 3.** Physiological and metabolic adaptation in bermudagrass against salt stress [36–40]. $H_2O_2$—hydrogen peroxide; EL—electrolyte leakage; MDA—malondialdehyde; Pn—net photosynthetic rate; Chl—chlorophyll; TNC—total non-structural carbohydrates; ABA—abscisic acid; SA—salicylic acid; JA—jasmonic acid; ROS—reactive oxygen species; GB—glycine betaine; SOD—superoxide dismutase; CAT—catalase; APX—ascorbate peroxidase; DHAR—dehydroascorbate reductase.

## 5. Anatomical Responses

Bermudagrass has salt glands on the adaxial surface of its leaves that respond to high salinity conditions. The density of salt glands is also enhanced by increasing salinity levels, but variation is observed in different bermudagrass cultivars. The presence of salt crystals on the surface of leaves, indicative of ion secretion, was investigated in bermudagrass [29]. They reported that salt glands become visible normally in the mesophyll tissue of bermudagrass (Figure 1). The activity of the exclusion of $Na^+$ shows the tolerance of a

particular species against salt (Figure 2) [6]. Salt glands are usually bicellular in the grass. In the family *Poaceae*, bicellular salt glands are present normally in the tribes Chlorideae, Sporoboleae, and Aeluropodeae [22]. Under salinity, the presence of $Na^+$ and $Cl^-$ in salt crystals adjacent to salt glands showed the activity of ion secretion [23,43]. The formation of glandular structures, which are specific mechanisms against salinity in bermudagrass cultivars, may result in higher levels of visible ion excretion. This confirmed that the rate of leaf salt excretion efficiency is a significant method of ion regulation that aids salinity tolerance [32]. Salt crystals were observed on both the adaxial and abaxial surfaces of the leaves of all species of bermudagrass subjected to salinity [44].

As the salt level increased, a corresponding decrease was observed in the diameter of the root tissue. Salinity is recognized to encourage the suberization of the root endodermis and hypodermis. Salinity-tolerant cultivars are often recognized by lignified walls of cortical parenchyma and the inner thicker tangential walls of endodermis tissues [37]. The parenchyma cells, both cortical and pith, are enhanced. These tissues boost the storage capacity, which is vital in unfavorable, moist environments [33]. In the salt-tolerant cultivars of *C. dactylon*, the stem area was noticeably enhanced under a saline regime. This enhanced succulence in the stem area, which might be helpful in the storage of additional water and nutrients, and finally, improved survival under harsh environments. Moreover, the increased thickness of sclerenchyma and a reduction in the size of sclerenchyma cells were noted in the stem cells of some populations. This quality may help confer resistance to water loss through the stem area and can play a vital part in adaptation to harsh conditions. Enlarged metaxylem and phloem areas in some populations possibly play a central role in the transportation of water and photosynthates, predominantly under severe saline conditions. In some bermudagrass cultivars, the increased cortical area under salt-stress conditions may be vital under physiological drought conditions for better water storage capacity. The epidermal cell area of stem tissue was increased with increasing salinity levels; a thick epidermis is an adaptive feature of salt-tolerant species [45,46].

Both the sieve tube and phloem areas are significantly enhanced in bermudagrass under salt-stress conditions (Figure 3). The stem area is enhanced by increased succulence and thickness of sclerenchyma tissues and epidermis for preventing water loss. There is increased thickness of the cortex for enhanced storage of water, a greater number of vascular bundles, protoxylem, metaxylem, phloem, and sieve areas for better transfer of photosynthates, water, and nutrients. However, xylem vessels remain highly stable [37].

## 6. Physiological and Metabolic Responses

The physiological mechanisms such as photosynthesis, transpiration rate, relative water content (RWC), leaf temperature, and osmotic potential are significantly affected under high salt conditions. The components of photosynthesis such as photosynthetic pigments, enzymes, thylakoid membrane lipids, and proteins are also disturbed under salt stress [43]. The enzymes protect the cellular membranes and structures from the harmful effects of free radicals [13]. The stability of the cellular membrane, the relative water content of the leaf, the transpiration rate, the chlorophyll content of the leaf, and the level of starch all are considerably decreased. However, leaf total photosynthetic rate (Pn) and stomatal conductance remain stable under saline conditions [36,47]. The content of chlorophyll significantly decreased in the leaves. The causes behind this decline might be the deprivation of chlorophyll content, and partially closed stomata under saline environmental conditions (Figure 3) [48]. Soluble sugars significantly increased in moderate salt environments but declined as the salt level increased. The starch content in shoots reduced with the increase in the salinity level. As mentioned above, salt-tolerant bermudagrass maintained significantly higher starch content in salty conditions by maintaining their turgor pressure and osmotic balance to stay alive.

Thus, one technique is transforming starch into simpler sugars [49]. Proline content is higher in the leaves under a salt-stress environment. Normally, salinity favors a reduction in photosynthesis activity; however, bermudagrass shows a significantly higher photosyn-

thetic rate. Electrolyte leakage is visibly increased in saline conditions, but in *C. dactylon*, the rate of this characteristic was visibly lower (Figure 3). Salt stress triggered a decline in the relative water content (RWC) of shoots, but it showed that it is more tolerant to salt than other grass species [50]. It was acknowledged that the leaves of salinity-tolerant bermudagrass genotypes hold a significantly higher quantity of organic osmotica (proline and total soluble proteins, total free amino acids, and total soluble sugars) than the salt-sensitive genotypes. This might facilitate osmotic adjustment for cellular turgor maintenance or counteract the osmotic stress from salt accumulation (Figure 3) [42].

Concentrations of total nonstructural carbohydrate (TNC) increased in the crown but remained unaffected in the roots when the salt condition increased (Figure 3). However, a small quantity of photosynthate might be recorded for an increase in root structural carbohydrates, but the storage of nonstructural carbohydrates in the crown portion and bases of the leaf was high. The survival and recovery of bermudagrass in a saline environment are through the transport of photosynthate from top to root by osmotic adjustment and redistribution. In addition, ionic substitution, the storage of carbohydrates, and increased levels of organic acids in the cell sap are also significant factors in survival [24].

The total photosynthesis rate, osmotic potential, leaf water potential, and turgor potential in bermudagrass are affected by increasing salt levels. While a reduction in osmotic potential and leaf water occurs, the net photosynthesis rate is not considerably affected by increased $NaCl$, $CaCl_2$, or $K_2SO_4$. The unpredictability in the total rate of photosynthesis might be due to the difficulty in precisely measuring leaf area and variation in blade and sheath proportions [19]. As a pigment of photosynthesis, chlorophyll is very susceptible to stressful environments. To examine the change in the composition of chlorophyll under salinity stress, the ratio of Chl a/b was calculated in bermudagrass, which was constant [51]. In all genotypes of grass, the total quantity of chlorophyll declined in salinity stress conditions but exhibited different magnitudes of reduction [7]. Under a control situation, high chlorophyll a and b production was observed in bermudagrass. However, the total chlorophyll content is the sum of chlorophyll a and b. The cultivars *C. dactylon* "Satiri" and *C. dactylon* "Tifdwarf" showed a higher quantity of total chlorophyll as compared to other cultivars [7,52].

## 7. Biochemical Responses

### 7.1. Antioxidant Enzyme System

The antioxidant enzymatic defense systems of grasses are weakened by salinity. High salt concentrations enhance reactive oxygen species (ROS), hydrogen peroxide ($H_2O_2$) content, and malondialdehyde (MDA) in the older leaves of salt-sensitive cultivars of bermudagrass (Figure 3) [53]. There is an up-regulation of genes used in the production of ascorbate peroxidase (APX), superoxide dismutase (SOD), and catalase (CAT), so the action of radical scavenging enzymes is enhanced significantly under salt stress [13]. Two types of antioxidants, enzymatic and non-enzymatic, detoxify ROS. Enzymatic antioxidants such as superoxide dismutase (SOD), catalase (CAT), and ascorbate peroxidase (APX) are greatly activated in tolerant cultivars of bermudagrass when exposed to salinity. Research findings have suggested that maintaining antioxidant enzyme activity plays a crucial role in cell elongation and expansion, mitosis, the synthesis of proteins and nucleotides, other cellular structures, and the expression of genes against stresses. Although the specific isoforms of antioxidant enzymes taking part in salinity tolerance might differ within different turfgrass cultivars, the intensity of salt tolerance still remains to be determined [54]. Research has established that levels of MDA and $H_2O_2$ in older leaves of the salt-sensitive genotypes were higher than in the salt-tolerant genotypes. However, SOD, APX, and CAT activities were much higher in the tolerant genotypes than in salt-sensitive ones, as in Figure 3 [13]. Different biotic (bacteria, fungi) and abiotic stresses (salt, drought, cold, heavy metals, pesticides) cause the over-production of ROS, damage to DNA, protein degradation, lipid peroxidation, and overall reduced crop yield [55]. The production of ROS causes membrane damage and ion leakage, but in response, cells have evolved a system to scavenge these

species. In the case of biotic stress, such as against heavy metal stress, SOD activates and manages oxidative stress and excessive ROS. The abiotic stresses cause water deficit conditions, so stomatal closure occurs to prevent further water loss and limits the $CO_2$ for photosynthesis. These conditions increase ROS, so plants activate CAT and SOD to protect against negative effects [56]. Peroxidase is present in mitochondria, peroxisomes, and Golgi apparatus functions in the removal of excessive $H_2O_2$ to protect the plant from the negative effects of ROS. The excessive production of $O_2$ and $H_2O_2$ as a result of photosynthesis and photorespiration is scavenged by SOD and CAT, respectively. Thus, POD is involved in fine regulation while CAT is involved in the mass scavenging of $H_2O_2$ [57,58]. The concentration of some enzymatic antioxidants against salt stress, such as glutathione reductase (GTR), polyphenol oxidase (PPO), and glutathione peroxidase (GPX), is not clear yet in bermudagrass. Therefore, under high salinity, the scavenging of ROS may be attributed to non-enzymatic antioxidants, including glutathione, $\alpha$-tocopherols, flavons, carotenoids, and ascorbic acid (AsA) [43].

*7.2. Osmolytes*

The harmful effects of salinity can be minimized by the accumulation of organic substances, such as proline, soluble sugars, soluble proteins, and glycine betaine (GB) (Figure 3) [49]. Proline is an amino acid more frequently found to accumulate more than 8 to 15 times in salinity-tolerant genotypes than in salinity-sensitive cultivars. Proline has two ends, hydrophilic and hydrophobic. The hydrophilic end binds to water while the hydrophobic end bind to proteins. Moreover, these protein molecules can bind with more water molecules, thus preventing protein denaturation under salt stress. It seems that proline accumulation reduces internal osmotic imbalances, acts as an antioxidant to scavenge ROS, and protects the enzymatic system and the functionalities of organelles under stress [43]. There are 12 different kinds of betaines in plants, but glycine betaine is the most widely found among them. An un-regulation of glycine betaine biosynthesis was observed among members of *Gramineae* under salt stress. Previous studies showed that it could help in many metabolic processes, the formation of plant alkaloids, and maintaining membrane integrity and enzyme activity [51]. Non-structural carbohydrates (glucose, sucrose, starch), polyols (inositol, mannitol), and soluble sugars are also produced under salt stress. In particular, sugars are involved in signaling mechanisms and responses to environmental stress, while carbohydrates are involved in plant development [59]. It has been found that the shoot length reduction and increased root length might be because of increased metabolites, the accumulation of soluble sugars, and significantly increased nitrogen metabolism. This physiological alteration could improve the root absorption capacity and tolerance to salt in bermudagrass [60].

## 8. Molecular and Proteomic Responses

Different modifications of morphological, physiological, and molecular processes are induced as a result of salt stress in the plant (Figure 3). However, these changes can vary notably between different genotypes or cultivars of *C. dactylon*. By increasing the levels of salt treatments, the antioxidant-related genes were down-regulated in older leaves but up-regulated in newer ones [13]. However, 77 types of proteins were found through a comparative analysis of proteomics under salt and drought stress in bermudagrass. A large number of these known proteins are involved in redox metabolic pathways, oxidative pentose phosphate, photosynthesis, and glycolysis. Thirteen of the known proteins were considered to be regulated properly by salt stress [51]. Most of them are involved in electron transport and energy pathways. Recently, four small RNA libraries recognized from cold-, mock-, salt-, or cold-plus-salt-treated *C. dactylon* genotypes, showed 449 miRNAs. Compared with the mock-treated genotypes and after the salt treatments, they found 49 miRNAs that were up-regulated and 39 that were down-regulated. Another study showed a total of 43 miRNAs and specific genes related to salt-stress responses [61]. They categorized the germplasms into four main groups based on leaf firing percentage and

relative shoot weight. In China, germplasms of various bermudagrass cultivars covering the range sensitive to tolerance against salinity stress were collected [62]. The research recognized four *C. dactylon* genotypes and showed that the most susceptible cultivar was Khabbal, whereas the most tolerant species was Tifway against salt stress [16]. With an in vitro selection method, for salinity tolerant calli, selections on solid medium have high salt concentrations. With this selection method, the scientists were confident in redeveloping several cultivars of *C. dactylon* with higher tolerance against salinity as compared to the parental genotype [63]. In addition, exogenously applied polyamine and mowing height in bermudagrass increased salinity tolerance [64,65].

Modern studies have revealed that various biotic and abiotic stresses, especially salt stress, also have a crucial role in *WRKY* genes [66]. In plants, WRKY transcription factors (TFs) comprise a very large family of genes, which could further be divided into three subcategories (I-III) based on the structure type of zinc fingers and the number of WRKY domains. Group I contains two WRKY domains, and groups II and III have one WRKY domain [67]. The multiple sequence alignment (MSA) of CdWRKY50 showed that the CdWRKY50 protein comprised a permanent WRKY domain, as WRKYGKK, followed by a C2H2-zinc-finger structure type on the N-terminus, which belonged to group II [68]. This indicated that CdWRKY50 belongs to a WRKY TF of *C. dactylon*.

Meanwhile, under NaCl treatment, the transcripts of two closely related WRKYs (AtWRKY25 and AtWRKY33) were up-regulated. The Atwrky33 and Atwrky25Atwrky33 mutants showed salt-sensitive phenotypes, while *AtWRKY25AtWRKY33* overexpression leads to enhanced salt tolerance in bermudagrass [69]. With the help of full-length transcriptome data, more than 100 WRKY TFs have been recognized in bermudagrass [70]. The CdWRKY50-silencing bermudagrass conferred enhanced salt tolerance. However, in an abiotic stress response, especially salt stress, the function of WRKY genes in bermudagrass is still under investigation. In a follow-up study, in wild *C. dactylon*, a salt-induced *CdWRKY50* was isolated and examined. The *CdWRKY50* gene expression was significantly affected by salt, abscisic acid (ABA) treatments, and cold. Its location was confirmed in the nucleus by subcellular localization analysis. Here, an increase in some oxidative stress-related genes was revealed in CdWRKY50-silencing in *C. dactylon* under salinity conditions [67]. Moreover, *AtGSTF8*, *AtGSTU19*, and *ChVDE* were up-regulated under salt stress [71]. Therefore, it can be considered that the inhibition of *CdWRKY50* boosts salt tolerance in bermudagrass to some extent through antioxidant activation to compensate for cell membrane damage (Table 1).

These findings indicate that in *C. dactylon*, salt stress is regulated by *CdWRKY50* by different pathways. Ultimately, it indicates that WRKY TFs have a member CdWRKY50, which is involved in the salt-stress responses of plants. It has been demonstrated that salt, drought, and ABA treatments up-regulated *CdWRKY50*. The tolerance of *C. dactylon* against salt stress was improved through the VIGS method by silencing the expression of *CdWRKY50* [67,69]. In salinity stress, the SOS signal pathway maintained the regulation of ionic homeostasis. It plays a vital role in the regulation of $Na^+/K^+$ homeostasis, and the genes activated are *SOS1*, *SOS2*, and *SOS3* [72]. Recent studies specified that in roots, $Na^+$ efflux, and in the xylem, the loading of $Na^+$ ions, are mediated by *SOS1*. *SOS3* enables the *SOS2–SOS3* complex formation, which is enabled by the perceived $Ca^{2+}$ signal, which then triggers phosphorylates and the movement of SOS1 transport [26].

NDPK (nucleoside diphosphate kinase), involved in responses to many abiotic stresses, is a metabolic enzyme [73–75] that is normally enhanced by Put (putrescine), Spd (spermidine), and Spm (spermine). Comparative proteomic analysis revealed that the higher NDPK protein level in *C. dactylon* significantly improved salt tolerance. The pretreatment of bermudagrass with exogenous polyamines showed more accumulation of proline and soluble sugars under salt-stress conditions as compared to untreated control plants by balancing osmotic pressure under stress situations. Generally, the regulation of 36 proteins was confirmed through comparative proteomic analysis in bermudagrass under salt stress. Among them, some proteins were involved in the electron transport chain, and ROS and

energy pathways were significantly improved [76–78]. Interestingly, in this study, some proteins were observed that were usually affected by polyamines, such as triosephosphate isomerase, fructose-bisphosphate aldolase, and glyceraldehyde-3-phosphate dehydrogenase A. These proteins were found in chloroplasts and involved in the photosystem's Calvin cycle. Some proteins are involved in the glycolysis process, such as cytoplasmic fructose, phosphate aldolase, and cytosolic triosephosphate isomerase. The remaining are involved in the gluconeogenese/glyoxylate cycle, such as "pyruvate orthophosphate dikinase" and "pyruvate, phosphate dikinase". Seven proteins involved in the carbon fixation process that are frequently regulated may result in a change in sugar content, which represents the polyamines-mediated Calvin cycle, glycolysis, and the gluconeogenese/glyoxylate cycle. These can play significant roles against salt responses in bermudagrass [64].

**Table 1.** Expression of genes against salinity stress in bermudagrass [67,78–82].

| Serial No | Gene Involved | Functions | References |
|---|---|---|---|
| 1 | *CdWRKY50* | Antioxidant activation, Cell membrane damage | [67] |
| 2 | *COR, LEA, POD-1* | Protect plant from damage under salt stress | [78] |
| 3 | *CdSOD1, CdPOD1, CdPOD2, CdCAT2* | Antioxidant activation, Oxidative stress | [79] |
| 4 | *psbA1, psbB1, psbP, psbY, ECA4, RAN1, MHX1* | Ion-homeostasis, photosynthesis-related | [80] |
| 5 | *BeDREB1, BeDREB2* | Role in signal transduction against salt stress | [81] |
| 6 | *Cdt-NY-YC1* | Osmotic stress, ion leakage | [82] |

Genes were up-regulated in pathways of protein translational modification such as ubiquitination and kinases. In particular, a large number of E3 RING and E3 SCF proteins and their associated genes were notably induced by salt stress, signifying that these enzymes may be involved in ways that may be regular on the 26S proteasome during the response to salt stress. The overexpression of the genes of antioxidant enzymes such as *CdSOD1, CdPOD1, CdPOD2, CdCAT1,* and *CdCAT2* was observed under salt stress [79]. It was established that the *LEA* gene was up-regulated under salt stress in bermudagrass species. CBF1 (C-repeat binding factors) transcription factor is a member of the *CBF* gene family. It can trigger the expression of a large subset of downstream (cold-regulated) *COR* and *POD-1* genes up-regulated under salinity conditions, thereby protecting plants from damage. In *C. dactylon*, to study the expression of gene profiles under salt stress, the expression levels of seven prominent genes, namely *ECA4, RAN1, MHX1, psbA1, psbB1, psbP,* and *psbY* were assessed (Table 1). The outcome showed that all these genes were up-regulated noticeably under varied treatments compared to the control [80]. Moreover, under high-salinity environments, it was confirmed that the expression of many genes such as *BeDREB1, BeDREB2* responsible for stress-related signal transduction was reported (Table 1) [81]. An elevated tolerance to salinity stress was observed in the expression of an ABA up-regulated gene *NF-YC* and *Cdt-NY-YC1*. Higher concentration was observed in roots and rhizomes than in the stems and leaves involved in osmotic stress, water loss, and ion leakage [82]. The mitogen-activated protein kinase (MAPK) cascades play a vital role in the response to salt stress by regulating the expression of salt-related genes. Genes that activate salt stress positively are *AtMEKK1***,** *AtMKK2***,** *GhMPK2, ZmMPK5,* and *GhMPK17* [83].

## 9. Conclusions and Future Perspectives

The current review discusses the morphological, physiological, biochemical, and molecular mechanisms of bermudagrass under salt stress. Salt stress leads to abnormal plant growth and development, wilting, water deficiency, the production of ROS, decreased respiration rate, cell membrane damage, and changes in amino acid composition. In the case of biochemical mechanisms, the activation of antioxidant enzymes (SOD, POD, and

CAT) and the production of primary and secondary metabolites (proline) in response to salt have been discussed. Structural modifications are increased salt gland production, the formation of salt crystals, enlarged phloem and metaxylem, and the enhanced growth of the shoots/crown area but reduced root growth.

The production of salinity-tolerant bermudagrass cultivars is the key to resolving the difficulty of sustaining *C. dactylon* growth in saline soil. Moreover, the genomic sequence of only some of the cultivars of *C. dactylon* has been published, so there is a need to explore other *Cynodon* cultivars possessing the potential to be genetically modified for improved tolerance to salt stress. In areas where there is a high risk of salinity, it is recommended that salt-resistant species be used. In addition, selection and breeding programs designed to improve this species' ability to cope with salt exposure should be feasible. Therefore, different approaches and products could be employed to reduce the salt effect on bermudagrass by regulating numerous genetic, hormonal, and metabolic pathways.

**Author Contributions:** Conceptualization: M.N., X.-B.Y. and S.-N.S.; writing and editing, M.N., X.-B.Y.; review J.-B.F., J.-X.Z., L.G. and C.-J.Z.; supervision, X.-B.Y. All authors have read and agreed to the published version of the manuscript.

**Funding:** This work was financially supported by the National Natural Science Foundation of China (Nos. 32171672 and 31702165) and the Project of Forestry Science and Technology Innovation and Promotion of Jiangsu (Grant No. LYKJ[2021]09).

**Institutional Review Board Statement:** Not applicable.

**Informed Consent Statement:** Not applicable.

**Data Availability Statement:** Not applicable.

**Conflicts of Interest:** The authors declare no conflict of interest.

## Abbreviations

| | |
|---|---|
| SOD | Superoxide dismutase |
| POD | Peroxide dismutase |
| APX | Ascorbate peroxidase |
| CAT | Catalase |
| DHAR | Dehydroascorbate reductase |
| PPO | Polyphenol oxidase |
| GPOX | Glutathione peroxidase |
| ROS | Reactive oxygen species |
| MDA | Malondialdehyde |
| SOS | Salt overly sensitive |
| NDPK | Nucleoside diphosphate kinase |
| LEA | Late embryogenesis abundant |
| MAPK | Mitogen-activated protein kinase |
| Put | Putrescine |
| Spd | Spermidine |
| Spm | Spermine |
| CBF | C-repeat binding factors |
| TF | Transcription factors |
| COR | Cold-regulated |

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
