# Peer review of "Bermudagrass Responses and Tolerance to Salt Stress by the Physiological, Molecular Mechanisms and Proteomic Perspectives of Salinity Adaptation"

_agronomy, doi:10.3390/agronomy13010174_

Round 1

Reviewer 1 Report (New Reviewer)

The review manuscript deals with morphological, anatomical, physiological, biochemical and molecular responses of bermudagrass against salinity stress. The paper presents state of the art of different plant reactions to salinity based on available references. The manuscript has logical workflow, however, there are some points which have to be corrected and expanded. The comments are listed below:

1. Introduction:

Add general mechanisms of other plants tolerance/adaptation to salty environment and mechanisms of halophytes adaptation to salty environment for comparison

L12: Poaceae in italics throughout the manuscript

L43: I suggest to indicate reference after the Figure in brackets throughout the paper: (Figure 1) [7].

L43: bermudagrass (Cynodon dactylon)

Physiological and metabolic responses:

L245-248: move this statement to point 7.1

Biochemical responses:

7.1. Antioxidant enzyme system

To underline the versatility of antioxidant enzymes and nonenzymatic antioxidants add general information about their role in biotic and different abiotic stresses mitigation (pesticides, heavy metals, drought), you can include following references:

https://doi.org/10.1007/s00425-022-03838-x
https://doi.org/10.1016/j.scienta.2022.110988
https://doi.org/10.1080/15569543.2018.1471091
https://doi.org/10.3390/plants10030436

Add some studies related to POD in alleviation of salinity stress in bermudagrass. POD is also one of the basic antioxidant enzyme examined in stress conditions.

8. Molecular and proteomic responses

L339: regulated in newer ones

L398: kinase)

L399: Put, Apd, and Spm – add full names

L400: C. dactylon in italics

L436: against salinity stress in bermudagrass

Conclusion and Future Perspectives:

write genus and species name in italics

Author Response

Manuscript ID:  Agronomy-2112549

Title: Bermudagrass Responses and Tolerance to Salt Stress by the Physiological, Molecular Mechanisms and Proteomic Perspectives of Salinity Adaptation

Responses to Reviewer’s comments

We are very grateful for the editor and anonymous reviewers for the received comments and valuable suggestions which let us improve our manuscript. Below you will see the point-by-point response to the reviewers’ comments. All the changes and corrections are made by track changes on. I hope it can now be suitable to be published in Agronomy.

Reviewer 1 Responses

  1. Introduction:

Add general mechanisms of other plants tolerance/adaptation to salty environment and mechanisms of halophytes adaptation to salty environment for comparison.

RESPONSE: Thanks for the guidance. The mechanisms of halophytes against salt stress has been added.

L12: Poaceae in italics throughout the manuscript

RESPONSE: Yes, accepted and corrected.

L43: I suggest to indicate reference after the Figure in brackets throughout the paper: (Figure 1) [7].

RESPONSE: Thanks for the correction. The reference at the end of every figure has been added.

L43: bermudagrass (Cynodon dactylon)

RESPONSE: Thanks. Accepted and added.

Physiological and metabolic responses:

L245-248: move this statement to point 7.1

RESPONSE: Thanks for the guidance. These lines have been added under heading 7.1.

Biochemical responses:

7.1. Antioxidant enzyme system

To underline the versatility of antioxidant enzymes and nonenzymatic antioxidants add general information about their role in biotic and different abiotic stresses mitigation (pesticides, heavy metals, drought).

RESPONSE: Thanks for the useful suggestions. The role of enzymatic and nonenzymatic antioxidants against biotic and abiotic stresses have been added briefly.

Add some studies related to POD in alleviation of salinity stress in bermudagrass. POD is also one of the basic antioxidant enzymes examined in stress conditions.

RESPONSE: Thanks for the guidance. The basic role of almost every antioxidant (POD, SOD, CAT) but especially role of POD under salt stress has been added.

  1. Molecular and proteomic responses

L339: regulated in newer ones

L398: kinase)

L399: Put, Apd, and Spm – add full names

L400: C. dactylon in italics

L436: against salinity stress in bermudagrass

RESPONSE: Thanks for the corrections. All the corrections in “molecular and proteomic” portion have been made according to your suggestions.

Conclusion and Future Perspectives:

write genus and species name in italics

RESPONSE: Yes, Accepted and corrected.

Reviewer 2 Report (Previous Reviewer 2)

The review titled (Bermudagrass Responses and Tolerance to Salt Stress by the Physiological, Molecular Mechanisms and Proteomic Perspectives of Salinity Adaptation) by Noor et al. explored bermudagrass's morphological, physiological, biochemical, molecular, and proteomic mechanisms under salt stress. The review tried to answer how bermudagrass grows and adapts well under salt conditions.
In my opinion, the authors made a lot of effort to improve the review; I think the text is much improved. However, there are minor errors in the text.
-    The resolution of all the figures must be improved.
-    Lines 56 to 58 (The study established that in Cynodon sp, the activity of salt exclusion was weaker; therefore, it was 57 highly salt tolerant as compared to other grasses) rephrase this sentence. It is not understandable
-    lines 202 to 204 are not suitable for this section
-    lines 207 to 215 are not suitable for this section
-    lines 270 to 272 are not suitable for this section
-    C. dactylon must be italic throughout the manuscript
Best Regards,

Author Response

Manuscript ID:  Agronomy-2112549

Title: Bermudagrass Responses and Tolerance to Salt Stress by the Physiological, Molecular Mechanisms and Proteomic Perspectives of Salinity Adaptation

Responses to Reviewer’s comments

We are very grateful for the editor and anonymous reviewers for the received comments and valuable suggestions which let us improve our manuscript. Below you will see the point-by-point response to the reviewers’ comments. All the changes and corrections are made by track changes on in the MS. I hope it can now be suitable to be published in Agronomy.

Reviewer 2 Response

In my opinion, the authors made a lot of effort to improve the review; I think the text is much improved. However, there are minor errors in the text.
The resolution of all the figures must be improved.

Lines 56 to 58 (The study established that in Cynodon sp, the activity of salt exclusion was weaker; therefore, it was 57 highly salt tolerant as compared to other grasses) rephrase this sentence. It is not understandable

RESPONSE: Thank you very much for the appreciation. According to your suggestions, figures are improved and the lines are revised to make them readable now.

-lines 202 to 204 are not suitable for this section.
-lines 207 to 215 are not suitable for this section.
-lines           270            to          272           are            not           suitable           for           this           section.

RESPONSE: Thanks for the corrections, Accepted and removed these lines from the manuscript.

  1. dactylon must be italic throughout the manuscript

RESPONSE: Yes, C. dactylon italicized throughout the manuscript.

Reviewer 3 Report (New Reviewer)

Dear Editor

This manuscript is a good work but needs some revisions before acceptance:

1- A native English-spoken person should check the language of the manuscript.

2- Some parts are not complete. For instance, biochemical and molecular responses need more references, and I have found more essential references in these two parts.

3- The Discussion part is not acceptable. The authors should improve the conclusion part.

4- It is better to add a comprehensive table for all the sections with references.

Author Response

Manuscript ID:  Agronomy-2112549

Title: Bermudagrass Responses and Tolerance to Salt Stress by the Physiological, Molecular Mechanisms and Proteomic Perspectives of Salinity Adaptation

Responses to Reviewer’s comments

We are very grateful for the editor and anonymous reviewers for the received comments and valuable suggestions which let us improve our manuscript. Below you will see the point-by-point response to the reviewers’ comments. All the changes and corrections are made by track changes on in the MS. I hope it can now be suitable to be published in Agronomy.

Reviewer 3 Response

Dear Editor

This manuscript is a good work but needs some revisions before acceptance:

1- A native English-spoken person should check the language of the manuscript.

RESPONSE: Thank you very much for the appreciation. We have revised it from a native English person to make it more readable.

2- Some parts are not complete. For instance, biochemical and molecular responses need more references, and I have found more essential references in these two parts.

RESPONSE: Thanks, we have added some new information in the biochemical and molecular portion as well.

3- The Discussion part is not acceptable. The authors should improve the conclusion part.

RESPONSE: Yes, Accepted and corrected.

4- It is better to add a comprehensive table for all the sections with references.

RESPONSE: Thanks, I have already added the figures for all the sections but their references were missing, so I have added their references now. A table for “Molecular and Proteomic” portion was mandatory as they have genes and their functions in it. So, I have drawn a table for it. Other headings haven’t such data for tables, So, that’s why their figures are drawn.  

Round 2

Reviewer 1 Report (New Reviewer)

The Authors have corrected the paper according to most of the suggestions. However, I have still some comments:

L332: heavy metals are abiotic stress. In the types of abiotic stress in brackets include also pesticides

L332-344: this is a big part of the text without references. Only two are given at the end of the text.

L334: and overall reduced crop yield - at the end of this statement add previously proposed references: https://doi.org/10.1007/s00425-022-03838-x

and https://doi.org/10.1080/15569543.2018.1471091

L337: remove: such as drought, salt and chilling. It was mentioned above

L340: and SOD to protect against negative effects - at the end of this statement some references should be added: https://doi.org/10.1016/j.scienta.2022.110988 and https://doi.org/10.3390/plants10030436

Author Response

Manuscript ID:  Agronomy-2112549

Title: Bermudagrass Responses and Tolerance to Salt Stress by the Physiological, Molecular Mechanisms and Proteomic Perspectives of Salinity Adaptation

Responses to Reviewer’s comments

We are very grateful for the editor and anonymous reviewers for the received comments and valuable suggestions which let us improve our manuscript. Below you will see the point-by-point response to the reviewers’ comments. All the changes and corrections are made by track changes on. I hope it can now be suitable to be published in Agronomy.

Reviewer 1 Responses

  1. L332: heavy metals are abiotic stress. In the types of abiotic stress in brackets include also pesticides

RESPONSE: Thanks for the correction. In the types of abiotic stress, pesticides have been added in line 302.

  1. L332-344: this is a big part of the text without references. Only two are given at the end of the text.

L334: and overall reduced crop yield - at the end of this statement add previously proposed references.

RESPONSE: Yes, accepted and corrected. According to your suggestion, new references have been added in line 303.

  1. L337: remove: such as drought, salt and chilling. It was mentioned above

RESPONSE: Accepted and corrected in L 306.

  1. L340: and SOD to protect against negative effects - at the end of this statement some references should be added

RESPONSE: Thanks. Accepted and added in L 309.

Reviewer 3 Report (New Reviewer)

Dear Editor

I hope this message finds you well.

The authors have corrected the manuscript and it is now acceptable for publication. Regards

Author Response

Thank you very much for your time. We are grateful for your appreciation and recommendation.

This manuscript is a resubmission of an earlier submission. The following is a list of the peer review reports and author responses from that submission.

Round 1

Reviewer 1 Report

Primarily, the work is full of statements without a single piece of data.

The paper cites known facts without a critical review.

In the text, you mention pictures that are not visible anywhere, eg Figures 1,2,4,5 .....20. I have the feeling that the paper was compiled from a part of the dissertation.

Figue 21., if you are talking about bermuda grass, at least put a picture of the grass, not dicotyledons. There should be more literature sources for a review article. Only 28 out of 88 literary sources are younger than 10 years. It should be supplemented with more recent literature. References is not written according to the guide for authors. For example years in references 1, 3, 20, etc. are not bolded. Headings under 22 and 19 are in capital letters. Edit double space in some parts; line 495-496.

Likewise, you did not touch on some enzymes whose role has not  been clarified "yet", for example PPO, GPOX or non enzymatic components like  phenolic acid, flavons  etc.

Author Response

Manuscript ID:  Agronomy-1977427

Title: Bermudagrass Responses and Tolerance to Salt Stress by the Physiological, Molecular Mechanisms and Proteomic Perspectives of Salinity Adaptation

Responses to Reviewer’s comments

We are very grateful for the editor and anonymous reviewers for the received comments and valuable suggestions which let us improve our manuscript. Below you will see the point-by-point response to the reviewers’ comments. All the changes and corrections are highlighted in yellow color. I hope it can now be suitable to be published in Agronomy.

Reviewer 1 Responses

In the text, you mention pictures that are not visible anywhere e g Figures 1,2,4,5 .....20. I have the feeling that the paper was compiled from a part of the dissertation.

RESPONSE: Thanks for the correction. More figures are added and suitably cited in the manuscript to make it understandable.

Figue 21., if you are talking about bermudagrass, at least put a picture of the grass, not dicotyledons. There should be more literature sources for a review article. Only 28 out of 88 literary sources are younger than 10 years. It should be supplemented with more recent literature. References is not written according to the guide for authors. For example years in references 1, 3, 20, etc. are not bolded. Headings under 22 and 19 are in capital letters. Edit double space in some parts; line 495-496.

RESPONSE: Yes, accepted and corrected. The picture of bermudagrass has been clicked by one of the author and added in Figure 1. The references younger than 10 years have been added and highlighted with yellow color, see reference no 1,9,12 and so on. Heading under 19 has been corrected and under 22 has been updated with recent reference according to author’s guide. Double spaces and comma mistakes are corrected after reading by two authors.

Likewise, you did not touch on some enzymes whose role has not been clarified "yet", for example PPO, GPOX or non enzymatic components like phenolic acid, flavons etc.

RESPONSE: Thanks for the useful suggestion; the role of some enzymes has not been clarified yet, in bermudagrass but studied in some other plants and grasses. So, we have added them according to your suggestions in line no 299-303.

Reviewer 2 Report

The review titled (Bermudagrass Responses and Tolerance to Salt Stress by the Physiological, Molecular Mechanisms and Proteomic Perspectives of Salinity Adaptation) by Noor et al.

This review explores bermudagrass's morphological, physiological, biochemical, molecular, and proteomic mechanisms under salt stress. The review tried to answer how bermudagrass grows and adapts well under salt conditions.

In my opinion, the review has fundamental errors in the text that reflect inaccuracy from the authors

1-The review needs English editing. Please find some suggested punctuation and English corrections in the attached file.

2- Line 29 (Therefore, the current study aims to …….) change study to review.

3-The authors must update the cited references; I noticed that there was not even one cited research in the last three years.

5-all the figures are very simple, and the authors need to be more creative; I had the impression that all the figures are repeated, and nothing novel in the figures; authors need to make figures that summarize mainly effects and adaptive characteristics of bermudagrass under salinity stress

6- All the figures' numbers and its citation in the text must be revised; for example, figure 3 must be figure 1. In several cases, the authors cite figures not found in the review.

7- Fundamentally, the authors need to include table(s) concluding all the studied or isolated genes and their relationship with salinity stress.

8- All genes' names must be in italic.

 9- paragraph 398 to 400 is not related to the studied title.

10- What is the relationship between section 10 and salinity tolerance? All the cited references do not investigate any salinity stress. Deleting it or citing research that explains your title is highly recommended.

Best Regards,

Author Response

Manuscript ID:  Agronomy-1977427

Title: Bermudagrass Responses and Tolerance to Salt Stress by the Physiological, Molecular Mechanisms and Proteomic Perspectives of Salinity Adaptation

Responses to Reviewer’s comments

We are very grateful for the editor and anonymous reviewers for the received comments and valuable suggestions which let us improve our manuscript. Below you will see the point-by-point response to the reviewers’ comments. All the changes and corrections are highlighted in yellow color. I hope it can now be suitable to be published in Agronomy.

Reviewer 2 Response

In my opinion, the review has fundamental errors in the text that reflect inaccuracy from the authors

1-The review needs English editing. Please find some suggested punctuation and English corrections in the attached file.

RESPONSE: Thank you very much for this help. According to your suggestions manuscript is revised.

2- Line 29 (Therefore, the current study aims to …….) change study to review.

RESPONSE: Yes, corrected in line no 29

3-The authors must update the cited references; I noticed that there was not even one cited research in the last three years.

RESPONSE: Thanks for the useful suggestion, The recent references from years 2018-2022 have been added.

5-all the figures are very simple, and the authors need to be more creative; I had the impression that all the figures are repeated, and nothing novel in the figures; authors need to make figures that summarize mainly effects and adaptive characteristics of bermudagrass under salinity stress

RESPONSE: Thanks for the correction. All figures have been revised

6- All the figures' numbers and its citation in the text must be revised; for example, figure 3 must be figure 1. In several cases, the authors cite figures not found in the review.

RESPONSE: Yes, accepted and revised according to your suggestions and cited according to suitable place in line no 56,67,79 and so on.

7- Fundamentally, the authors need to include table(s) concluding all the studied or isolated genes and their relationship with salinity stress.

RESPONSE: Accepted, a table with all the isolated genes and their relationship with salt stress has been added.

8- All genes' names must be in italic.

RESPONSE: Yes, corrected in line no 339, 372, 416, 422 and 429.

 9- paragraph 398 to 400 is not related to the studied title.

RESPONSE: Yes, accepted and deleted.

10- What is the relationship between section 10 and salinity tolerance? All the cited references do not investigate any salinity stress. Deleting it or citing research that explains your title is highly recommended.

RESPONSE: Thanks for the correction. There is no relation of section 10 with the title so, we have deleted it.

Round 2

Reviewer 1 Report

When you submit your paper for publication, technical errors cannot happen like you did. IN THE PAPER, ONLY CITE THE OVERVIEW OF THE RESULTS OF OTHER RESEARCH. NO SCIENTIFIC CONTRIBUTION FROM YOU.

Reviewer 2 Report

It is evident that the authors don't pay much attention to the simplest rules of scientific writing. In the first round, I recommended revising all the numbers of the figures, but unfortunately, the authors keep the same mistakes. The authors insist on citing in the text figure numbers that it does not exist.

In addition to the comments relating to the science being reported, there are significant concerns about the manuscript's grammar, usage, and overall readability. Therefore, revise the text to fix grammatical errors and improve the text's overall readability. I suggest you have a fluent, preferably native, English-language speaker thoroughly copyedit your manuscript for language usage, spelling, and grammar. If you do not know anyone who can do this, MDPI can provide language editing services.

I am sorry to inform you that even after your effort in revising this review, it is not suitable to publish in an agronomy journal.